# Compressive spectral embedding: sidestepping the SVD

**Dinesh Ramasamy**
dineshr@ece.ucsb.edu
ECE Department, UC Santa Barbara

**Upamanyu Madhow**
madhow@ece.ucsb.edu
ECE Department, UC Santa Barbara

## Abstract

Spectral embedding based on the Singular Value Decomposition (SVD) is a widely used "preprocessing" step in many learning tasks, typically leading to dimensionality reduction by projecting onto a number of dominant singular vectors and rescaling the coordinate axes (by a predefined function of the singular value). However, the number of such vectors required to capture problem structure grows with problem size, and even partial SVD computation becomes a bottleneck. In this paper, we propose a low-complexity *compressive* spectral embedding algorithm, which employs random projections and finite order polynomial expansions to compute approximations to SVD-based embedding. For an $m \times n$ matrix with $\mathcal{T}$ non-zeros, its time complexity is $O\left((\mathcal{T} + m + n)\log(m+n)\right)$, and the embedding dimension is $O(\log(m+n))$, both of which are independent of the number of singular vectors whose effect we wish to capture. To the best of our knowledge, this is the first work to circumvent this dependence on the number of singular vectors for general SVD-based embeddings. The key to sidestepping the SVD is the observation that, for downstream inference tasks such as clustering and classification, we are only interested in using the resulting embedding to evaluate pairwise similarity metrics derived from the $\ell_2$-norm, rather than capturing the effect of the underlying matrix on arbitrary vectors as a partial SVD tries to do. Our numerical results on network datasets demonstrate the efficacy of the proposed method, and motivate further exploration of its application to large-scale inference tasks.

## 1 Introduction

Inference tasks encountered in natural language processing, graph inference and manifold learning employ the singular value decomposition (SVD) as a first step to reduce dimensionality while retaining *useful* structure in the input. Such spectral embeddings go under various guises: Principle Component Analysis (PCA), Latent Semantic Indexing (natural language processing), Kernel Principal Component Analysis, commute time and diffusion embeddings of graphs, to name a few. In this paper, we present a *compressive* approach for accomplishing SVD-based dimensionality reduction, or embedding, without actually performing the computationally expensive SVD step.

The setting is as follows. The input is represented in matrix form. This matrix could represent the adjacency matrix or the Laplacian of a graph, the probability transition matrix of a random walker on the graph, a bag-of-words representation of documents, the action of a kernel on a set of $l$ points $\{\mathbf{x}(p) \in \mathbb{R}^d : p = 1, \ldots, m\}$ (kernel PCA)[1][2] such as

$$A(p,q) = e^{-\|\mathbf{x}(p)-\mathbf{x}(q)\|^2 / 2\alpha^2} \quad \text{(or) } A(p,q) = I(\|\mathbf{x}(p) - \mathbf{x}(q)\| < \alpha), \ 1 \leq p, q \leq l, \quad (1)$$

where $I(\cdot)$ denotes the indicator function or matrices derived from $K$-nearest-neighbor graphs constructed from $\{\mathbf{x}(p)\}$. We wish to compute a transformation of the rows of this $m \times n$ matrix $A$ which succinctly captures the global structure of $A$ via *euclidean* distances (or similarity metrics derived from the $\ell_2$-norm, such as normalized correlations). A common approach is to com-

pute a partial SVD of $A$, $\sum_{l=1}^{l=k} \sigma_l \mathbf{u}_l \mathbf{v}_l^T$, $k \ll n$, and to use it to embed the rows of $A$ into a $k$-dimensional space using the rows of $E = [f(\sigma_1)\mathbf{u}_1 \ f(\sigma_2)\mathbf{u}_2 \ \cdots \ f(\sigma_k)\mathbf{u}_k]$, for some function $f(\cdot)$. The embedding of the variable corresponding to the $l$-th row of the matrix $A$ is the $l$-th row of $E$. For example, $f(x) = x$ corresponds to Principal Component Analysis (PCA): the $k$-dimensional rows of $E$ are projections of the $n$-dimensional rows of $A$ along the first $k$ principal components, $\{\mathbf{v}_l, \ l = 1, \ldots, k\}$. Other important choices include $f(x) = $ constant used to cut graphs [3] and $f(x) = 1/\sqrt{1-x}$ for commute time embedding of graphs [4]. Inference tasks such as (unsupervised) clustering and (supervised) classification are performed using $\ell_2$-based pairwise similarity metrics on the embedded coordinates (rows of $E$) instead of the ambient data (rows of $A$).

Beyond the obvious benefit of dimensionality reduction from $n$ to $k$, embeddings derived from the *leading* partial-SVD can often be interpreted as denoising, since the "noise" in matrices arising from real-world data manifests itself via the smaller singular vectors of $A$ (e.g., see [5], which analyzes graph adjacency matrices). This is often cited as a motivation for choosing PCA over "isotropic" dimensionality reduction techniques such as random embeddings, which, under the setting of the Johnson-Lindenstrauss (JL) lemma, can also preserve structure.

The number of singular vectors $k$ needed to capture the structure of an $m \times n$ matrix grows with its size, and two bottlenecks emerge as we scale: (a) The computational effort required to extract a large number of singular vectors using conventional iterative methods such as Lanczos or simultaneous iteration or approximate algorithms like Nystrom [6], [7] and Randomized SVD [8] for computation of partial SVD becomes prohibitive (scaling as $\Omega(k\mathcal{T})$, where $\mathcal{T}$ is the number of non-zeros in $A$) (b) the resulting $k$-dimensional embedding becomes unwieldy for use in subsequent inference steps.

**Approach and Contributions:** In this paper, we tackle these scalability bottlenecks by focusing on what embeddings are actually used for: computing $\ell_2$-based *pairwise* similarity metrics typically used for supervised or unsupervised learning. For example, K-means clustering uses pairwise Euclidean distances, and SVM-based classification uses pairwise inner products. We therefore ask the following question: "Is it possible to compute an embedding which captures the pairwise euclidean distances between the rows of the spectral embedding $E = [f(\sigma_1)\mathbf{u}_1 \ \cdots \ f(\sigma_k)\mathbf{u}_k]$, while sidestepping the computationally expensive partial SVD?" We answer this question in the affirmative by presenting a *compressive* algorithm which directly computes a low-dimensional embedding.

There are two key insights that drive our algorithm:
• By approximating $f(\sigma)$ by a low-order ($L \ll \min\{m, n\}$) polynomial, we can compute the embedding iteratively using matrix-vector products of the form $A\mathbf{q}$ or $A^T\mathbf{q}$.
• The iterations can be computed *compressively:* by virtue of the celebrated JL lemma, the embedding geometry is approximately captured by a small number $d = O(\log(m + n))$ of randomly picked starting vectors.

The number of passes over $A$, $A^T$ and time complexity of the algorithm are $L$, $L$ and $O(L(\mathcal{T} + m + n) \log(m + n))$ respectively. These are all *independent* of the number of singular vectors $k$ whose effect we wish to capture via the embedding. This is in stark contrast to embedding directly based on the partial SVD. Our algorithm lends itself to parallel implementation as a sequence of $2L$ matrix-vector products interlaced with vector additions, run in parallel across $d = O(\log(m + n))$ randomly chosen starting vectors. This approach significantly reduces both computational complexity and embedding dimensionality relative to partial SVD. A freely downloadable Python implementation of the proposed algorithm that exploits this inherent parallelism can be found in [9].

## 2 Related work

As discussed in Section 3.1, the concept of compressive measurements forms a key ingredient in our algorithm, and is based on the JL lemma [10]. The latter, which provides probabilistic guarantees on approximate preservation of the Euclidean geometry for a finite collection of points under random projections, forms the basis for many other applications, such as compressive sensing [11].

We now mention a few techniques for exact and approximate SVD computation, before discussing algorithms that sidestep the SVD as we do. The time complexity of the full SVD of an $m \times n$ matrix is $O(mn^2)$ (for $m > n$). Partial SVDs are computed using iterative methods for eigen decompositions of symmetric matrices derived from $A$ such as $AA^T$ and $[0 \ A^T; A \ 0]$ [12]. The

complexity of standard iterative eigensolvers such as simultaneous iteration[13] and the Lanczos method scales as $\Omega(k\mathcal{T})$ [12], where $\mathcal{T}$ denotes the number of non-zeros of $A$.

The leading $k$ singular value, vector triplets $\{(\sigma_l, \mathbf{u}_l, \mathbf{v}_l), \ l = 1, \ldots, k\}$ minimize the matrix *reconstruction* error under a rank $k$ constraint: they are a solution to the optimization problem $\arg\min \|A - \sum_{l=1}^{l=k} \sigma_l \mathbf{u}_l \mathbf{v}_l^T\|_F^2$, where $\|\cdot\|_F$ denotes the Frobenius norm. Approximate SVD algorithms strive to reduce this error while also placing constraints on the computational budget and/or the number of passes over $A$. A commonly employed approximate eigendecomposition algorithm is the Nystrom method [6], [7] based on random sampling of $s$ columns of $A$, which has time complexity $O(ksn + s^3)$. A number of variants of the Nystrom method for kernel matrices like (1) have been proposed in the literature. These aim to improve accuracy using preprocessing steps such as $K$-means clustering [14] or random projection trees [15]. Methods to reduce the complexity of the Nystrom algorithm to $O(ksn + k^3)$[16], [17] enable Nystrom sketches that see more columns of $A$. The complexity of all of these grow as $\Omega(ksn)$. Other randomized algorithms, involving iterative computations, include the Randomized SVD [8]. Since all of these algorithms set out to recover $k$-leading eigenvectors (exact or otherwise), their complexity scales as $\Omega(k\mathcal{T})$.

We now turn to algorithms that sidestep SVD computation. In [18], [19], vertices of a graph are embedded based on diffusion of probability mass in random walks on the graph, using the power iteration run independently on random starting vectors, and stopping "prior to convergence." While this approach is specialized to probability transition matrices (unlike our general framework) and does not provide explicit control on the nature of the embedding as we do, a feature in common with the present paper is that the time complexity of the algorithm and the dimensionality of the resulting embedding are independent of the number of eigenvectors $k$ captured by it. A parallel implementation of this algorithm was considered in [20]; similar parallelization directly applies to our algorithm. Another specific application that falls within our general framework is the commute time embedding on a graph, based on the normalized adjacency matrix and weighing function $f(x) = 1/\sqrt{1-x}$ [4], [21]. Approximate commute time embeddings have been computed using Spielman-Teng solvers [22], [23] and the JL lemma in [24]. The complexity of the latter algorithm and the dimensionality of the resulting embedding are comparable to ours, but the method is specially designed for the normalized adjacency matrix and the weighing function $f(x) = 1/\sqrt{1-x}$. Our more general framework would, for example, provide the flexibility of suppressing small eigenvectors from contributing to the embedding (e.g, by setting $f(x) = I(x > \epsilon)/\sqrt{1-x}$).

Thus, while randomized projections are extensively used in the embedding literature, to the best of our knowledge, the present paper is the first to develop a general compressive framework for spectral embeddings derived from the SVD. It is interesting to note that methods similar to ours have been used in a different context, to estimate the *empirical distribution* of eigenvalues of a large hermitian matrix [25], [26]. These methods use a polynomial approximation of indicator functions $f(\lambda) = I(a \leq \lambda \leq b)$ and random projections to compute an approximate histogram of the number of eigenvectors across different bands of the spectrum: $[a, b] \subseteq [\lambda_{\min}, \lambda_{\max}]$.

## 3  Algorithm

We first present the algorithm for a symmetric $n \times n$ matrix $S$. Later, in Section 3.5, we show how to handle a general $m \times n$ matrix by considering a related $(m + n) \times (m + n)$ symmetric matrix. Let $\lambda_l$ denote the eigenvalues of $S$ sorted in descending order and $\mathbf{v}_l$ their corresponding unit-norm eigenvectors (chosen to be orthogonal in case of repeated eigenvalues). For any function $g(x) : \mathbb{R} \mapsto \mathbb{R}$, we denote by $g(S)$ the $n \times n$ symmetric matrix $g(S) = \sum_{l=1}^{l=n} g(\lambda_l) \mathbf{v}_l \mathbf{v}_l^T$. We now develop an $O(n \log n)$ algorithm to compute a $d = O(\log n)$ dimensional embedding which approximately captures pairwise euclidean distances between the rows of the embedding $E = [f(\lambda_1) \mathbf{v}_1 \ f(\lambda_2) \mathbf{v}_2 \ \cdots \ f(\lambda_n) \mathbf{v}_n]$.

*Rotations are inconsequential:* We first observe that rotation of basis does not alter $\ell_2$-based similarity metrics. Since $V = [\mathbf{v}_1 \cdots \mathbf{v}_n]$ satisfies $VV^T = V^TV = \mathbb{I}_n$, pairwise distances between the rows of $E$ are equal to corresponding pairwise distances between the rows of $EV^T = \sum_{l=1}^{l=n} f(\lambda_l) \mathbf{v}_l \mathbf{v}_l^T = f(S)$. We use this observation to compute embeddings of the rows of $f(S)$ rather than those of $E$.

## 3.1 Compressive embedding

Suppose now that we know $f(S)$. This constitutes an $n$-dimensional embedding, and similarity queries between two "vertices" (we refer to the variables corresponding to rows of $S$ as vertices, as we would for matrices derived from graphs) requires $O(n)$ operations. However, we can reduce this time to $O(\log n)$ by using the JL lemma, which informs us that pairwise distances can be approximately captured by compressive projection onto $d = O(\log n)$ dimensions.

Specifically, for $d > (4 + 2\beta) \log n / (\epsilon^2/2 - \epsilon^3/3)$, let $\Omega$ denote an $n \times d$ matrix with i.i.d. entries drawn uniformly at random from $\{\pm 1/\sqrt{d}\}$. According to the JL lemma, pairwise distances between the rows of $f(S)\Omega$ approximate pairwise distances between the rows of $f(S)$ with high probability. In particular, the following statement holds with probability at least $1 - n^{-\beta}$: $(1 - \epsilon) \|\mathbf{u} - \mathbf{v}\|^2 \leq \|(\mathbf{u} - \mathbf{v})\Omega\|^2 \leq (1 + \epsilon) \|\mathbf{u} - \mathbf{v}\|^2$, for any two rows $\mathbf{u}, \mathbf{v}$ of $f(S)$.

The key take-aways are that (a) we can reduce the embedding dimension to $d = O(\log n)$, since we are only interested in pairwise similarity measures, and (b) We do not need to compute $f(S)$. We only need to compute $f(S)\Omega$. We now discuss how to accomplish the latter efficiently.

## 3.2 Polynomial approximation of embedding

Direct computation of $E' = f(S)\Omega$ from the eigenvectors and eigenvalues of $S$, as $f(S) = \sum f(\lambda_l)\mathbf{v}_l\mathbf{v}_l^T$ would suggest, is expensive ($O(n^3)$). However, we now observe that computation of $\psi(S)\Omega$ is easy when $\psi(\cdot)$ is a polynomial. In this case, $\psi(S) = \sum_{p=0}^{p=L} b_p S^p$ for some $b_p \in \mathbb{R}$, so that $\psi(S)\Omega$ can be computed as a sequence of $L$ matrix-vector products interlaced with vector additions run in *parallel* for each of the $d$ columns of $\Omega$. Therefore, they only require $Ld\mathcal{T} + O(Ldn)$ flops. Our strategy is to approximate $E' = f(S)\Omega$ by $\widetilde{E} = \widetilde{f}_L(S)\Omega$, where $\widetilde{f}_L(x)$ is an $L$-th order polynomial approximation of $f(x)$. We defer the details of computing a "good" polynomial approximation to Section 3.4. For now, we assume that one such approximation $\widetilde{f}_L(\cdot)$ is available and give bounds on the loss in fidelity as a result of this approximation.

## 3.3 Performance guarantees

The spectral norm of the "error matrix" $Z = f(S) - \widetilde{f}(S) = \sum_{r=1}^{r=n} (f(\lambda_r) - \widetilde{f}_L(\lambda_r))\mathbf{v}_r\mathbf{v}_r^T$ satisfies $\|Z\| = \delta = \max_l |f(\lambda_l) - \widetilde{f}_L(\lambda_l)| \leq \max\{|f(x) - \widetilde{f}_L(x)|\}$, where the spectral norm of a matrix $B$, denoted by $\|B\|$ refers to the induced $\ell_2$-norm. For symmetric matrices, $\|B\| \leq \alpha \iff |\lambda_l| \leq \alpha \; \forall l$, where $\lambda_l$ are the eigenvalues of $B$. Letting $\mathbf{i}_p$ denote the unit vector along the $p$-th coordinate of $\mathbb{R}^n$, the distance between the $p, q$-th rows of $\widetilde{f}(S)$ can be written as

$$\|\widetilde{f}_L(S)(\mathbf{i}_p - \mathbf{i}_q)\| = \|f(S)(\mathbf{i}_p - \mathbf{i}_q) - Z(\mathbf{i}_p - \mathbf{i}_q)\| \leq \|E^T(\mathbf{i}_p - \mathbf{i}_q)\| + \delta\sqrt{2}. \qquad (2)$$

Similarly, we have that $\|\widetilde{f}_L(S)(\mathbf{i}_p - \mathbf{i}_q)\| \geq \|E^T(\mathbf{i}_p - \mathbf{i}_q)\| - \delta\sqrt{2}$. Thus pairwise distances between the rows of $\widetilde{f}_L(S)$ approximate those between the rows of $E$. However, the distortion term $\delta\sqrt{2}$ is *additive* and must be controlled by carefully choosing $\widetilde{f}_L(\cdot)$, as discussed in Section 4.

Applying the JL lemma [10] to the rows of $\widetilde{f}_L(S)$, we have that when $d > O(\epsilon^{-2} \log n)$ with i.i.d. entries drawn uniformly at random from $\{\pm 1/\sqrt{d}\}$, the embedding $\widetilde{E} = \widetilde{f}_L(S)\Omega$ captures pairwise distances between the rows of $\widetilde{f}_L(S)$ up to a multiplicative distortion of $1 \pm \epsilon$ with high probability:

$$\left\|\widetilde{E}^T(\mathbf{i}_p - \mathbf{i}_q)\right\| = \left\|\Omega^T\widetilde{f}_L(S)(\mathbf{i}_p - \mathbf{i}_q)\right\| \leq \sqrt{1 + \epsilon}\left\|\widetilde{f}_L(S)(\mathbf{i}_p - \mathbf{i}_q)\right\|$$

Using (2), we can show that $\|\widetilde{E}^T(\mathbf{i}_p - \mathbf{i}_q)\| \leq \sqrt{1 + \epsilon}\left(\|E^T(\mathbf{i}_p - \mathbf{i}_q)\| + \delta\sqrt{2}\right)$. Similarly, $\|\widetilde{E}^T(\mathbf{i}_p - \mathbf{i}_q)\| \geq \sqrt{1 - \epsilon}\left(\|E^T(\mathbf{i}_p - \mathbf{i}_q)\| - \delta\sqrt{2}\right)$. We state this result in Theorem 1.

**Theorem 1.** *Let $\widetilde{f}_L(x)$ denote an $L$-th order polynomial such that: $\delta = \max_l |f(\lambda_l) - \widetilde{f}_L(\lambda_l)| \leq \max|f(x) - \widetilde{f}_L(x)|$ and $\Omega$ an $n \times d$ matrix with entries drawn independently and uniformly at random from $\{\pm 1/\sqrt{d}\}$, where $d$ is an integer satisfying $d > (4 + 2\beta) \log n / (\epsilon^2/2 - \epsilon^3/3)$. Let*

$g : \mathbb{R}^p \to \mathbb{R}^d$ *denote the mapping from the i-th row of* $E = [f(\lambda_1)\mathbf{v}_1 \cdots f(\lambda_n)\mathbf{v}_n]$ *to the i-th row of* $\widetilde{E} = \widetilde{f}_L(S)\Omega$. *The following statement is true with probability at least* $1 - n^{-\beta}$:

$$\sqrt{1-\epsilon}(\|u-v\| - \delta\sqrt{2}) \leq \|g(u) - g(v)\| \leq \sqrt{1+\epsilon}(\|u-v\| + \delta\sqrt{2})$$

*for any two rows* $u, v$ *of* $E$. *Furthermore, there exists an algorithm to compute each of the* $d = O(\log n)$ *columns of* $\widetilde{E}$ *in* $O(L(\mathcal{T} + n))$ *flops independent of its other columns which makes* $L$ *passes over* $S$ *(* $\mathcal{T}$ *is the number of non-zeros in* $S$*).*

### 3.4 Choosing the polynomial approximation

We restrict attention to matrices which satisfy $\|S\| \leq 1$, which implies that $|\lambda_l| \leq 1$. We observe that we can trivially center and scale the spectrum of any matrix to satisfy this assumption when we have the following bounds: $\lambda_l \leq \sigma_{\max}$ and $\lambda_l \geq \sigma_{\min}$ via the rescaling and centering operation given by: $S' = 2S/(\sigma_{\max} - \sigma_{\min}) - (\sigma_{\max} + \sigma_{\min})\mathbb{I}_n/(\sigma_{\max} - \sigma_{\min})$ and by modifying $f(x)$ to $f'(x) = f(x(\sigma_{\max} - \sigma_{\min})/2 + (\sigma_{\max} + \sigma_{\min})/2)$.

In order to compute a polynomial approximation of $f(x)$, we need to define the notion of "good" approximation. We showed in Section 3.3 that the errors introduced by the polynomial approximation can be summarized by furnishing a bound on the spectral norm of the error matrix $Z = f(S) - \widetilde{f}_L(S)$: Since $\|Z\| = \delta = \max_l |f(\lambda_l) - \widetilde{f}_L(\lambda_l)|$, what matters is how well we approximate the function $f(\cdot)$ *at* the eigenvalues $\{\lambda_l\}$ of $S$. Indeed, if we know the eigenvalues, we can minimize $\|Z\|$ by minimizing $\max_l |f(\lambda_l) - \widetilde{f}_L(\lambda_l)|$. This is not a particularly useful approach, since computing the eigenvalues is expensive. However, we can use our prior knowledge of the domain from which the matrix $S$ comes from to penalize deviations from $f(\lambda)$ differently for different values of $\lambda$. For example, if we know the distribution $p(x)$ of the eigenvalues of $S$, we can minimize the average error $\Delta_L = \int_{-1}^{1} p(\lambda)|f(\lambda) - \widetilde{f}_L(\lambda)|^2 \mathrm{d}x$. In our examples, for the sake of concreteness, we assume that the eigenvalues are uniformly distributed over $[-1, 1]$ and give a procedure to compute an $L$-th order polynomial approximation of $f(x)$ that minimizes $\Delta_L = (1/2)\int_{-1}^{1} |f(x) - \widetilde{f}_L(x)|^2 \mathrm{d}x$.

A numerically stable procedure to generate finite order polynomial approximations of a function over $[-1, 1]$ with the objective of minimizing $\int_{-1}^{1} |f(x) - \widetilde{f}_L(x)|^2 \mathrm{d}x$ is via Legendre polynomials $p(r, x)$, $r = 0, 1, \ldots, L$. They satisfy the recursion $p(r, x) = (2 - 1/r)xp(r-1, x) - (1 - 1/r)p(r-2, x)$ and are orthogonal: $\int_{-1}^{1} p(k, x)p(l, x)\mathrm{d}x = 2I(k = l)/(2r + 1)$. Therefore we set $\widetilde{f}_L(x) = \sum_{r=0}^{r=L} a(r)p(r, x)$ where $a(r) = (r + 1/2)\int_{-1}^{1} p(r, x)f(x)\mathrm{d}x$. We give a method in Algorithm 1 that uses the Legendre recursion to compute $p(r, S)\Omega$, $r = 0, 1, \ldots, L$ using $Ld$ matrix-vector products and vector additions. The coefficients $a(r)$ are used to compute $\widetilde{f}_L(S)\Omega$ by adding weighted versions of $p(r, S)\Omega$.

---

**Algorithm 1** Proposed algorithm to compute approximate $d$-dimensional eigenvector *embedding* of a $n \times n$ symmetric matrix $S$ (such that $\|S\| \leq 1$) using the $n \times d$ random projection matrix $\Omega$.

---

1: **Procedure** FASTEMBEDEIG($S, f(x), L, \Omega$):
2: //* Compute polynomial approximation $\widetilde{f}_L(x)$ which minimizes $\int_{-1}^{1} |f(x) - \widetilde{f}_L(x)|^2 \mathrm{d}x$ *//
3: **for** $r = 0, \ldots, L$ **do**
4:    $a(r) \leftarrow (r + 1/2)\int_{x=-1}^{x=1} f(x)p(r, x)\mathrm{d}x$    //* $p(r, x)$: Order $r$ Legendre polynomial *//

5: $Q(0) \leftarrow \Omega, Q(-1) \leftarrow 0, \widetilde{E} \leftarrow a(0)Q(0)$
6: **for** $r = 1, 2, \ldots, L$ **do**
7:    $Q(r) \leftarrow (2 - 1/r)SQ(r-1) - (1 - 1/r)Q(r-2)$    //* $Q(r) = p(r, S)\Omega$ *//
8:    $\widetilde{E} \leftarrow \widetilde{E} + a(r)Q(r)$    //* $\widetilde{E}$ now holds $\widetilde{f}_r(S)\Omega$ *//

9: **return** $\widetilde{E}$    //* $\widetilde{E} = \widetilde{f}_L(S)\Omega$ *//

---

As described in Section 4, if we have prior knowledge of the distribution of eigenvalues (as we do for many commonly encountered large matrices), then we can "boost" the performance of the generic Algorithm 1 based on the assumption of eigenvalues uniformly distributed over $[-1, 1]$.

### 3.5 Embedding general matrices

We complete the algorithm description by generalizing to any $m \times n$ matrix $A$ (not necessarily symmetric) such that $\|A\| \leq 1$. The approach is to utilize Algorithm 1 to compute an approximate $d$-dimensional embedding of the symmetric matrix $S = [0 \ A^T; A \ 0]$. Let $\{(\sigma_l, \mathbf{u}_l, \mathbf{v}_l) : l = 1, \ldots, \min\{m,n\}\}$ be an SVD of $A = \sum_l \sigma_l \mathbf{u}_l \mathbf{v}_l^T$ ($\|A\| \leq 1 \iff \sigma_l \leq 1$). Consider the following spectral mapping of the rows of $A$ to the rows of $E_{\text{row}} = [f(\sigma_1)\mathbf{u}_1 \ \cdots \ f(\sigma_m)\mathbf{u}_m]$ and the columns of $A$ to the rows of $E_{\text{col}} = [f(\sigma_1)\mathbf{v}_1 \ \cdots \ f(\sigma_n)\mathbf{v}_n]$. It can be shown that the unit-norm orthogonal eigenvectors of $S$ take the form $[\mathbf{v}_l; \mathbf{u}_l]/\sqrt{2}$ and $[\mathbf{v}_l; -\mathbf{u}_l]/\sqrt{2}$, $l = 1, \ldots, \min\{m,n\}$, and their corresponding eigenvalues are $\sigma_l$ and $-\sigma_l$ respectively. The remaining $|m - n|$ eigenvalues of $S$ are equal to 0. Therefore, we call $\widetilde{E}_{\text{all}} \leftarrow$ FASTEMBEDEIG$(S, f'(x), L, \Omega)$ with $f'(x) = f(x)I(x \geq 0) - f(-x)I(x < 0)$ and $\Omega$ is an $(m + n) \times d, d = O(\log(m + n))$ matrix (entries drawn independently and uniformly at random from $\{\pm 1/\sqrt{d}\}$). Let $\widetilde{E}_{\text{col}}$ and $\widetilde{E}_{\text{row}}$ denote the first $n$ and last $m$ rows of $\widetilde{E}_{\text{all}}$. From Theorem 1, we know that, with overwhelming probability, pairwise distances between any two rows of $\widetilde{E}_{\text{row}}$ approximates those between corresponding rows of $E_{\text{row}}$. Similarly, pairwise distances between any two rows of $\widetilde{E}_{\text{col}}$ approximates those between corresponding rows of $E_{\text{col}}$.

## 4 Implementation considerations

We now briefly go over implementation considerations before presenting numerical results in Section 5.

**Spectral norm estimates** In order to ensure that the eigenvalues of $S$ are within $[-1, 1]$ as we have assumed, we scale the matrix by its spectral norm ($\|S\| = \max|\lambda_l|$). To this end, we obtain a tight lower bound (and a good approximation) on the spectral norm using power iteration (20 iterates on $6 \log n$ randomly chosen starting vectors), and then scale this up by a small factor (1.01) for our estimate (typically an upper bound) for $\|S\|$.

**Polynomial approximation order** $L$: The error in approximating $f(\lambda)$ by $\widetilde{f}_L(\lambda)$, as measured by $\Delta_L = \int_{-1}^{1} |f(x) - \widetilde{f}_L(x)|^2 dx$ is a non-increasing function of the polynomial order $L$. Reduction in $\Delta_L$ often corresponds to a reduction in $\delta$ that appears as a bound on distortion in Theorem 1. "Smooth" functions generally admit a lower order approximation for the same target error $\Delta_L$, and hence yield considerable savings in algorithm complexity, which scales linearly with $L$.

**Polynomial approximation method**: The rate at which $\delta$ decreases as we increase $L$ depends on the function $p(\lambda)$ used to compute $\widetilde{f}_L(\lambda)$ (by minimizing $\Delta_L = \int p(\lambda)|f(\lambda) - \widetilde{f}_L(\lambda)|^2 dx$). The choice $p(\lambda) \propto 1$ yields the Legendre recursion used in Algorithm 1, whereas $p(\lambda) \propto 1/\sqrt{1 - \lambda^2}$ corresponds to the Chebyshev recursion, which is known to result in fast convergence. We defer to future work a detailed study of the impact of alternative choices for $p(\lambda)$ on $\delta$.

**Denoising by cascading** In large-scale problems, it may be necessary to drive the contribution from certain singular vectors to zero. In many settings, singular vectors with smaller singular values correspond to noise. The number of such singular values can scale as fast as $O(\min\{m, n\})$. Therefore, when we place nulls (zeros) in $f(\lambda)$, it is desirable to ensure that these nulls are *pronounced* after we approximate $f(\lambda)$ by $\widetilde{f}_L(\lambda)$. We do this by computing $(\widetilde{g}_{L/b}(S))^b \Omega$, where $\widetilde{g}_{L/b}(\lambda)$ is an $L/b$-th order approximation of $g(\lambda) = f^{1/b}(\lambda)$. The small values in the polynomial approximation of $f^{1/b}(\lambda)$ which correspond to $f(\lambda) = 0$ (nulls which we have set) get amplified when we pass them through the $x^b$ non-linearity.

## 5 Numerical results

While the proposed approach is particularly useful for large problems in which exact eigendecomposition is computationally infeasible, for the purpose of comparison, our results are restricted to smaller settings where the exact solution can be computed. We compute the exact partial eigendecomposition using the ARPACK library (called from MATLAB). For a given choice of weighing

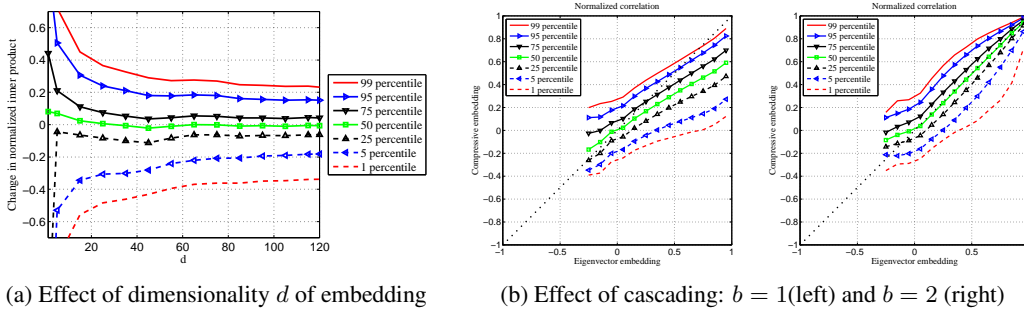

| (a) Effect of dimensionality $d$ of embedding | (b) Effect of cascading: $b = 1$(left) and $b = 2$ (right) |

Figure 1: DBLP collaboration network normalized correlations

function $f(\lambda)$, the associated embedding $E = [f(\lambda_1)\mathbf{v}_1 \; \cdots \; f(\lambda_n)\mathbf{v}_n]$ is compared with the compressive embedding $\widetilde{E}$ returned by Algorithm 1. The latter was implemented in Python using the Scipy's sparse matrix-multiplication routines and is available for download from [9].

We consider two real world undirected graphs in [27] for our evaluation, and compute embeddings for the normalized adjacency matrix $\widetilde{A}$ ($= D^{-1/2}AD^{-1/2}$, where $D$ is a diagonal matrix with row sums of the adjacency matrix $A$; the eigenvalues of $\widetilde{A}$ lie in $[-1, 1]$) for graphs. We study the accuracy of embeddings by comparing pairwise normalized correlations between $i, j$-th rows of $E$ given by $< E(i,:), E(j,:) >/\|E(i,:)\|\|E(j,:)\|$ with those predicted by the approximate embedding $< \widetilde{E}(i,:), \widetilde{E}(j,:) > /\|\widetilde{E}(i,:)\|\|\widetilde{E}(j,:)\|$ ($E(i,:)$ is short-hand for the $i$-th row of $E$).

**DBLP collaboration network** [27] is an undirected graph on $n = 317080$ vertices with 1049866 edges. We compute the leading 500 eigenvectors of the normalized adjacency matrix $\widetilde{A}$. The smallest of the five hundred eigenvalues is 0.98, so we set $f(\lambda) = I(\lambda \geq 0.98)$ and $S = \widetilde{A}$ in Algorithm 1 and compare the resulting embedding $\widetilde{E}$ with $E = [\mathbf{v}_1 \; \cdots \; \mathbf{v}_{500}]$. We demonstrate the dependence of the quality of the embedding $\widetilde{E}$ returned by the proposed algorithm on two parameters: (i) number of random starting vectors $d$, which gives the dimensionality of the embedding and (ii) the boosting/cascading parameter $b$ using this dataset.

*Dependence on the number of random projections $d$:* In Figure (1a), $d$ ranges from 1 to $120 \approx 9 \log n$ and plot the 1-st, 5-th, 25-th, 50-th, 75-th, 95-th and 99-th percentile values of the deviation between the compressive normalized correlation (from the rows of $\widetilde{E}$) and the corresponding exact normalized correlation (rows of $E$). The deviation decreases with increasing $d$, corresponding to $\ell_2$-norm concentration (JL lemma), but this payoff saturates for large values of $d$ as polynomial approximation errors start to dominate. From the 5-th and 95-th percentile curves, we see that a significant fraction (90%) of pairwise normalized correlations in $\widetilde{E}$ lie within $\pm 0.2$ of their corresponding values in $E$ when $d = 80 \approx 6 \log n$. For Figure (1a), we use $L = 180$ matrix-vector products for each randomly picked starting vector and set cascading parameter $b = 2$ for the algorithm in Section 4.

*Dependence on cascading parameter $b$:* In Section 4 we described how cascading can help suppress the contribution to the embedding $\widetilde{E}$ of the eigenvectors whose eigenvalues lie in regions where we have set $f(\lambda) = 0$. We illustrate the importance of this boosting procedure by comparing the quality of the embedding $\widetilde{E}$ for $b = 1$ and $b = 2$ (keeping the other parameters of the algorithm in Section 4 fixed: $L = 180$ matrix-vector products for each of $d = 80$ randomly picked starting vectors). We report the results in Figure (1b) where we plot percentile values of compressive normalized correlation (from the rows of $\widetilde{E}$) for different values of the exact normalized correlation (rows of $E$). For $b = 1$, the polynomial approximation of $f(\lambda)$ does not suppress small eigenvectors. As a result, we notice a deviation (bias) of the 50-percentile curve (green) from the ideal $y = x$ dotted line drawn (Figure 1b left). This disappears for $b = 2$ (Figure 1b right).

The running time for our algorithm on a standard workstation was about two orders of magnitude smaller than partial SVD using off-the-shelf sparse eigensolvers (e.g., the 80 dimensional embedding of the leading 500 eigenvectors of the DBLP graph took 1 minute whereas their exact computation

took 105 minutes). A more detailed comparison of running times is beyond the scope of this paper, but it is clear that the promised gains in computational complexity are realized in practice.

**Application to graph clustering for the Amazon co-purchasing network** [27] : This is an undirected graph on $n = 334863$ vertices with $925872$ edges. We illustrate the potential downstream benefits of our algorithm by applying $K$-means clustering on embeddings (exact and compressive) of this network. For the purpose of our comparisons, we compute the first 500 eigenvectors for $\widetilde{A}$ explicitly using an exact eigensolver, and use an 80-dimensional compressive embedding $\widetilde{E}$ which captures the effect of these, with $f(\lambda) = I(\lambda \geq \lambda_{500})$, where $\lambda_{500}$ is the 500th eigenvalue. We compare this against the usual spectral embedding using the first 80 eigenvectors of $\widetilde{A}$: $E = [\mathbf{v}_1 \cdots \mathbf{v}_{80}]$. We keep the dimension fixed at 80 in the comparison because $K$-means complexity scales linearly with it, and quickly becomes the bottleneck. Indeed, our ability to embed a large number of eigenvectors directly into a low dimensional space ($d \approx 6 \log n$) has the added benefit of dimensionality reduction within the subspace of interest (in this case the largest 500 eigenvectors).

We consider 25 instances of $K$-means clustering with $K = 200$ throughout, reporting the median of a commonly used graph clustering score, modularity [28] (larger values translate to better clustering solutions). The median modularity for clustering based on our embedding $\widetilde{E}$ is $0.87$. This is significantly better than that for $E$, which yields median modularity of $0.835$. In addition, the computational cost for $\widetilde{E}$ is one-fifth that for $E$ (1.5 minutes versus 10 minutes). When we replace the exact eigenvector embedding $E$ with approximate eigendecomposition using Randomized SVD [8] (parameters: power iterates $q = 5$ and excess dimensionality $l = 10$), the time taken reduces from 10 minutes to 17 seconds, but this comes at the expense of inference quality: median modularity drops to $0.748$. On the other hand, the median modularity increases to $0.845$ when we consider exact partial SVD embedding with 120 eigenvectors. This indicates that our compressive embedding yields better clustering quality because it is able to concisely capture more eigenvectors(500 in this example, compared to 80 and 120 with conventional partial SVD). It is worth pointing out that, even for known eigenvectors, the number of dominant eigenvectors $k$ that yields the best *inference performance* is often unknown *a priori*, and is treated as a hyper-parameter. For compressive spectral embedding $\widetilde{E}$, an elegant approach for implicitly optimizing over $k$ is to use the embedding function $f(\lambda) = I(\lambda \geq c)$, with $c$ as a hyper-parameter.

# 6   Conclusion

We have shown that random projections and polynomial expansions provide a powerful approach for spectral embedding of large matrices: for an $m \times n$ matrix $A$, our $O((\mathcal{T} + m + n) \log(m + n))$ algorithm computes an $O(\log(m+n))$-dimensional compressive embedding that provably approximates pairwise distances between points in the desired spectral embedding. Numerical results for several real-world data sets show that our method provides good approximations for embeddings based on partial SVD, while incurring much lower complexity. Moreover, our method can also approximate spectral embeddings which depend on the entire SVD, since its complexity does not depend on the number of dominant vectors whose effect we wish to model. A glimpse of this potential is provided by the example of $K$-means based clustering for estimating sparse-cuts of the Amazon graph, where our method yields much better performance (using graph metrics) than a partial SVD with significantly higher complexity. This motivates further investigation into applications of this approach for improving downstream inference tasks in a variety of large-scale problems.

## Acknowledgments
This work is supported in part by DARPA GRAPHS (BAA-12-01) and by Systems on Nanoscale Information fabriCs (SONIC), one of the six SRC STARnet Centers, sponsored by MARCO and DARPA. Any opinions, findings, and conclusions or recommendations expressed in this material are those of the authors and do not necessarily reflect the views of the funding agencies.

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
