[Reviews · NeurIPS 2015]

Submitted by Assigned_Reviewer_1

A new embedding method is proposed by using ideas from compressed sensing literature which does not require SVD. The manuscript provides a good case for the intuition and back it up with analysis. Experiments on two datasets are provided.

This method gives access to the raw embedding which many alternate methods don't. However, it is not clear whether the resulting embedding has nice orthogonal structure like SVD or not? This property is not crucial for the applications that require pairwise distances for inference but is crucial to applications with more involved analysis.

Comparing the quality of results with true SVD results is a good experiment. But independent experiments showing running time of the proposed algorithm for large scale problems would have helped the reviewer to appreciate the proposed method. Probably comparisons with few other methods which also side-step SVD? Reviewer understand that other methods may not have same flexibility as the proposed method offer but one of the crucial factor here is time.

The manuscript is well written in terms of providing intuition and insight for the proposed algorithm. However in its current form, manuscript has a lot of typos and also notations are not consistent. It is very confusing at places whether the A is a data matrix i.e., rows are data instances and cols are the feature representation in high dimension or A is pairwise distances between instances eq(1).

SVD operation on large matrices is a bottleneck in various large scale analyses. A new method which avoid SVD but still provide access to the embedding and not just pairwise distances is a significant contribution.

Summary: The paper proposes a new method for spectral embedding and dimensionality reduction by avoiding the expensive SVD. In particular, proposed method develop the embedding algorithm by using concepts from compressed sensing literature -- finite order polynomial expansions of matrices. The paper is well written and describe the algorithm at length along with performance analysis w.r.t various parameters.

Submitted by Assigned_Reviewer_2

The authors propose a novel algorithm for computing "compressive embeddings" of a matrix, which is most often done via approximate partial SVD. They provide detailed motivation and background for the problem, reviewing applications in graph processing, NLP, etc. The proposed algorithm "side-steps" the SVD by leveraging the JL lemma to compute approximate pair-wise distances and polynomial approximations to incorporate approximate functions of the singular values (e.g., thresholds). Some implementation details are highlighted.

Overall, I think this a very good paper that presents simple, fast, effective, and (to my knowledge) novel algorithm for computing compressive embeddings, which is likely to be superior to Randomized SVD and pure random projections for many applications. The presentation is also relatively clear, if dense, despite a few minor errors and formatting issues (outlined below).

One issue is that the authors do not flesh out a detailed story of how the performance changes relative to the embedding dimension and types of polynomials that might be approximated. This is difficult due to space-constraints, but especially for the former it would have been helpful to have a plot relating accuracy to dimension (not just a comparison btw two dimension sizes) that essentially provided an empirical view of theorem 1. Moreover, since the top 500 eigenvalues were computed the \delta could be reasonably approximated and so an empirical vs. theoretical expectation plot is possible.

Related to the above, more concrete recommendations from the authors w.r.t. how to choose embedding dimensions and eigenvalue thresholds would be helpful. And it is somewhat misleading that the authors set the polynomial approximations given knowledge of the first 500 eigenvalues. Discussion of what to do when such prior computations are infeasible would be helpful.

Minor issues:

-\cite{foo}\cite{bar} -> \cite{foo,bar} everywhere - line 73: point (a) is unclear - line 113: iteration\cite{foo} -> iteration \cite{foo} - line 237: "of good" -> "of a good" - line 340: "(=" -> "=" - capitalize Euclidean, Hermitian etc.

Summary: A well-written, compelling paper. The proposed algorithm is insightful and likely to have an impact.

Submitted by Assigned_Reviewer_3

The authors aim at various interesting machine learning tasks. Avoiding expensive SVD is the main selling point of the paper. As the authors claim that the computational complexity of the algorithm is O(min(m,n)(T+m+n)log(m+n)), parallel computation improves the cost to O((T+m+n)log(m+n)). I am curious about how parallel SVD would do. Maybe a careful comparison with sparse parallel SVD is useful to convince readers the superiority. I am also curious the computational cost of Legenche polynomial when choosing the polynomial in the algorithm.

In the experiment section, it is surprising to me that changing d dramatically doesn't affect the result much. Is there an intuition why this is happening?

Some minor points are (1) A few typos are found in the paper, especially in the experiment section.

(2) Notations can be more precise. For example, the definition of X and f(A). Although I understand what the authors mean, it could be confusing for readers who are not familiar with these topics.

It would be nice if the authors could address these questions.
Summary: The paper computes embedding of matrix without using SVD to perform spectral methods. Theoretical bounds on projection dimension and polynomial approximation is provided. However, I am not sure about the originality of the paper. Improvements on the experiment section is needed to convince me the algorithm is advantageous.

Author Feedback
Author rebuttal: REVIEWER 1

- Orthogonality is exactly what we sacrifice in our compressive representation (e.g., we used d = 40 & 80 dimensional embeddings to capture up to K = 500 eigenvectors in our example).

- On comparisons with other algorithms: We have compared our algorithm with exact SVD and the recently proposed approximate Randomized SVD algorithm. Randomized SVD is an element-by-element approximation, and retains more of the noise spectrum (small eigenvalues), thus degrading inference quality. Except for Spielman and Teng's theoretical work on commute time embeddings (references [21]-[22]), we are not aware of any other work on spectral embeddings without computing the spectrum itself. However, the Spielman-Teng algorithm is restricted to normalized adjacency matrices with a specific weighting function, so that our algorithm represents a significant generalization.

- We realize reusing the symbol A to represent symmetric matrices (eigenvector embeddings) and feature vector matrices (typically not symmetric; SVD embeddings) can cause confusion. The construction of the input matrix depends on the modeler: it can be a symmetric matrix of pairwise similarities constructed from feature vectors (sparsified by truncating small values), or the feature vectors themselves stacked into a matrix. For the latter, the modification in Section 3.5 can be used to compute SVD embeddings. We will make sure we carefully distinguish these cases in the revised paper.

We will review the manuscript closely for typos.

REVIEWER 2

- To illustrate performance dependence on embedding dimension d, we will replace Figure 1(a) with one plotting the maximum & average distortion of pairwise distances vs d. CLT-based intuition (see response to Reviewer 3) indicates that distortion should scale as 1/sqrt(d), but the JL lemma based estimates in Theorem 1 are conservative. Comprehensive study of such scaling falls beyond our present scope.

- Regarding the impact of polynomial approximations, they apply to any spectral weighting function, but the smoother the function, the smaller the order L needed for a target \delta. Complexity grows as O(L(T+m+n) log(m+n)), hence smooth functions lead to considerable savings.

- Regarding knowledge of the top 500 eigenvalues used in our example, this is not required in practice. Our example was to illustrate that we could capture a desired number of eigenvalues accurately. We chose the top 500 because they could be computed based on a reasonable time budget.

- In practice, we would not know how many eigenvectors are needed. We recommend choosing embedding dimension and eigenvalue threshold based on problem structure (e.g., for PCA, we can set the threshold based on noise variance), or based on downstream inference performance. Increasing the embedding dimension incrementally until such criteria are met is easy: since each dimension is independent, we can add dimensions simply by running the algorithm on additional random starting vectors.

- If we do need an approximation based on the first k eigenvectors, we can do so by setting the corresponding eigenvalue threshold using the CDF of eigenvalues, estimated using techniques similar to ours in [24, 25].

We will summarize the preceding discussion in the revised paper, and also fix the typos pointed out by the reviewer (thank you for your careful reading).

REVIEWER 3

- Actually, we claim that the algorithm complexity is O(L(T+m+n)log(m+n)). This can be reduced to O(L(T+m+n)) with parallelism where L is the order of the polynomial approximation of f().

- Parallelism for speeding up matrix vector products applies to our algorithm and to existing SVD techniques in equal measure (and we use it in the Python implementation of our algorithm that we plan to release). In terms of space, exact SVD methods (even those for sparse matrices) cannot save beyond min(m,n) * number of eigs, since that is the size of the output they report. For the same reason they cannot have complexity smaller than O(num of eigs * min(m,n)), as we note in the paper. In our running time result comparison, we do use sparse parallel eigensolver routines (available via MATLAB).

- Our algorithm's complexity scales linearly with the order L of the polynomial approximation of the spectral weighting function. Smoother functions can be approximated by lower order polynomials, and therefore require less computation to approximate. The computations in the algorithm exploit the recursive structure of the Legendre poloynomials.

- Rough intuition on the effect of the embedding dimension d can be obtained using the Central Limit Theorem: Each squared distance follows the Gaussian(0, 1/d) distribution (across multiple runs of this randomized algorithm) and we therefore see that distortion scales as 1/sqrt(d). Since correlations are linearly related to distance-squares, we expect them to also scale as 1/sqrt{d} (i.e., a relatively slow decay with d).